# Mimicking Evolution with Reinforcement Learning

## Abstract

In nature, there are two processes driving the development of the brain: evolution
and learning. Evolution acts slowly, across generations, and amongst other things,
it defines what agents learn by changing their internal reward function. Learning
acts fast, within one's lifetime, and it quickly updates agents' policies to maximise
the evolved reward function. Although previous work has emulated both of these
processes working in tandem, the optimisation of the reward function in order to
serve the aims of the evolutionary process is very computationally expensive. This
work proposes a fixed reward function, the evolutionary reward, that aims to max-
imise the number of current (and future) genetically similar agents. Furthermore,
we propose a way to approximate the joint action value by averaging the action
values of other agents weighted by their genetic similarity. In a finite environment
with limited resources this techniques drives improved survival mechanisms and
reproductive success. Given that this reward function is fixed, we avoid the com-
putationally intense process of optimising it. We demonstrate the viability of our
evolutionary reward by testing it in two bio-inspired, open-ended environments and
monitoring a number of metrics such as population size and life expectancy. We
compare our technique with the state-of-the-art evolutionary algorithm: CMA-ES,
and show the superiority of work at producing agents that maximise the number of
its genes across time.

## 1 Introduction

Evolution is the only process we know of today that has given rise to general intelligence (as demon-
strated in animals, and specifically in humans). This fact has been inspiring artificial intelligence (AI)
researchers to run evolution in artificial worlds that mimic key properties of life on Earth. One of
these key properties is open-endedness. This means that, as in nature, the fitness function (or any
goal function) of the environment is not defined anywhere but it simply emerges from the survival
and reproduction of genes. For this reason, we call these environments *open-ended* evolutionary
environments (OEEE). They are never-ending environments where adaptable agents are competing
for a common limited-resource to survive and replicate their genes. Using them for research is the
focus of the field of artificial life (ALife).

Our ability to run evolution efficiently in OEEE will dictate the success of ALife. In this work
we speed up the way evolution is ran in OEEE by introducing Evolution via Evolutionary Reward
(EvER). In EvER, each agent is born with an evolutionary reward that, when maximised by a learning,
it also maximises the survival and reproduction of the agent's genes. Due to this property we say that
this reward is aligned with evolution. This allows learning to search for policies with increasingly
evolutionary fitness. Also, by guarantying this alignment we don't need to go through the expensive
process of aligning the agents' reward functions through evolution. This reward function was designed
to work on any OEEE.

Submitted to 35th Conference on Neural Information Processing Systems (NeurIPS 2021). Do not distribute.

In the remaining part of this introduction we 1) describe how evolution changes what we learn; 2) introduce our contribution and describe how maximising a reward function can lead to the maximisation of evolutionary fitness.

## 1.1 Evolving what to learn

In nature, there are two different mechanisms driving the development of the brain. Evolution acts slowly, across generations, and amongst other things, it defines what agents learn by changing their internal reward function. Learning acts fast, within one's lifetime, and it quickly updates agents' policies to maximise pleasure and minimise pain. Combining these two methods has a long history in AI research [1, 42, 8]. This combination (illustrated in Appendix B, Figure 3) results in a very computationally expensive algorithm as it requires two loops 1) learning (the inner loop) where agents maximise their innate reward functions across their lifetimes and 2) evolution (the outer loop) where natural selection and mutation defines the reward functions for the next generation (amongst other things, such as NN topologies and initial weights).

We say that a reward function is aligned with evolution when the maximisation of the reward leads to the maximisation of the agent's fitness. Through evolution the most aligned reward functions get selected and increase their numbers. Intuitively, one can define the optimally aligned reward function as the reward function that allows a learner to learn most quickly how to maximise its fitness, assuming the conditions of the world (including other agents) remain the same. However, as agents evolve and learn, they change their environment and its corresponding fitness function. This change, increases the misalignment between the reward and fitness functions. Therefore, the optimally aligned reward function is always chasing the ever changing fitness function (see Appendix C for a formal description of this). However, in this paper, we show that in simulation it is possible to *define* a fixed reward function which is always aligned, although not guaranteed to be optimally aligned, with the essence of fitness: the ability of the individual to survive and reproduce its genes.

Our work allows learning to single-handedly drive the search for policies with increasingly evolutionary fitness by ensuring the alignment of the reward function with the fitness function. This greatly simplifies the two-loop algorithm used to combine evolution and learning that was described earlier in this section. We can do this because our reward is extrinsic to the agent and therefore, only possible within a simulation.

## 1.2 Learning to maximise evolutionary fitness

The distinction between an agent and a gene is key to understanding this paper. Formally, evolution is a change in gene frequencies in a population (of agents) over time. The gene is the unit of evolution, and an agent carries one or more genes. Richard Dawkins has famously described our bodies as throwaway survival machines built for replicating immortal genes [6]. His line illustrates well the gene-centered view of evolution [43, 6], a view that has been able to explain multiple phenomena such as intragenomic conflict and altruism that are difficult to explain with organism-centered or group-centered viewpoints [2, 10, 7]. From the gene's perspective, the evolutionary process is a constant competition for resources. However, from the agent's perspective, the evolutionary process is a mix between a cooperative exercise with agents that carry some of its genes (its family) and a competition with unrelated agents. Evolution pressures agents to engage in various degrees of collaboration depending on the degree of kinship between them and the agents they interact with (i.e. depending on the amount of overlap between the genes they carry). This pressure for cooperation amongst relatives was named *kin selection* [34].

Evolution acts on the gene level, but RL acts on the agent level. RL can be aligned with the evolutionary process by noting what evolution does to the agents through its selection of genes: evolution generates agents with increasing capabilities to maximise the survival and reproduction success of the genes they carry.

## 2   Related work

**Combining evolution and learning**   Combining evolution and learning has long history in AI research. The evolutionary reinforcement learning algorithm, introduced in 1991 [1], makes the evolutionary process determine the initial weights of two neural networks: an action and an evaluation

network. During an agent's lifetime, learning adapts the action network guided by the output of its innate and fixed (during its lifetime) evaluation network. NEAT+Q [42] uses an evolutionary algorithm, NEAT [36], to evolve topologies of NN and their initial weights so that they can better learn using RL. In NEAT-Q the reward function remains fixed. However, evolutionary algorithms have also been used to evolve potential-based shaping rewards and meta-parameters for RL [8].

**Competing in Arms-race**   Every time adaptable entities compete against each other an arms-race is created. Each entity's task gets harder every time their competitors learn something useful. This arms race drives the continued emergence of ever new innovative and sophisticated capabilities necessary to out-compete adversaries. Evolutionary Algorithms (EA) have been successfully used to co-evolve multiple competing entities [32, 29]. However, in sequential decision problems EA algorithms discard most of the information by not looking at the whole state-action trajectories the agents encounter throughout their lifetime. This theoretical disadvantage limits their potential efficiency to tackle sequential problems when compared with RL. Empirically, EA algorithms usually have a higher variance when compared with gradient methods [30, 23, 24]. With regards to gradient methods (deep learning methods in particular), impressive results have been recently achieved by training NN, through back-propagation, to compete against each other in simulated games (OpenFive [4], AlphaZero [31], GAN [11]). More closely aligned with our proposed methodology, OpenAI has recently developed Neural MMO [37], a simulated environment that captures some important properties of life on Earth. In Neural MMO artificial agents, represented by NN, need to forage for food and water to survive in a never-ending simulation. Currently, Neural MMO agents can not reproduce and their goal is to maximise their own survival, instead of maximising the survival and reproduction success of their genes as it happens in nature. We extend this work by introducing genes, the ability for agents to reproduce and we align the agents' reward with evolution. These are key properties of life on Earth that we must have in simulation environments if we hope to have them evolve similar solutions to the ones evolved by nature (in other words, these are key properties to achieve convergent evolution - see Appendix **??** for more details on why this important for AI research).

**Cooperative MARL**   Cooperative MARL is an active research area within RL that has been experiencing fast progress [26, 3, 9]. The setting is usually approached in a binary way [4, 41, 20]. Agents are grouped into teams and agents within the same team fully cooperate amongst each other whilst agents from different teams don't cooperate at all (cooperation is either one or zero); we define this scenario as the binary cooperative setting. The teams may have a fixed number of members or change dynamically [19, 27, 40, 5]. The most straightforward solution for this setting would be to train independent learners to maximise their team's reward. However, independent learners would face a non-stationary learning problem. The MADDPG [22] algorithm tackles this problem by using a multi-agent policy gradient method with a centralised critic and decentralised actors so that training takes into account all the states and actions of the entire team but during execution each agent can act independently. More relevant to our work, factored value functions[12, 27] such as Transfer Planning [40] Value Decomposition Networks (VDN) [38] and Q-Mix [28] use different methods to decompose the team's central action-value function into the decentralised action-value functions. We build on top of VDN (which is further explained in the Appendix D) to extend the concept of team to the concept of family and introduce continuous degrees of cooperation.

## 3   Background

**Reinforcement Learning**   We recall the single agent fully-observable RL setting [39], where the environment is typically formulated as a Markov decision process (MDP). At every time step, $t = 1, 2, \ldots$, the agent observes the environment's state $s_t \in \mathcal{S}$, and uses it to select an action $a_t \in \mathcal{A}$. As a consequence, the agent receives a reward $r_t \in \mathcal{R} \subset \mathbb{R}$ and the environment transitions to the state $s_{t+1}$. The tuple $(s_{t+1}, r_t)$ is sampled from the static probability distribution $p : \mathcal{S} \times \mathcal{A} \to \mathcal{P}(\mathcal{S} \times \mathcal{R})$ whilst the actions $a_t$ are sampled from the parametric policy function $\pi_\theta : \mathcal{S} \to \mathcal{P}(\mathcal{A})$:

$$s_{t+1}, r_t \sim p(s_{t+1}, r_t | s_t, a_t), \quad a_t \sim \pi_\theta(a_t | s_t) \tag{1}$$

The goal of the agent is to find the optimal policy parameters $\theta^*$ that maximise the expected return $\bar{R} = \mathbb{E}[\sum_{t=0}^{\infty} \gamma^t r_t]$, where $\gamma$ is the discount factor. In the more general framework, the state is only partially observable, meaning that the agent can not directly observe the state but instead it

observes $o_t \in \mathcal{O}$ which is typically given by a function of the state. In this situation, the environment is modelled by a partial observable Markov decision process (POMDP) and the policy usually incorporates past history $h_t = a_0 o_0 r_0, \ldots, a_{t-1} o_{t-1} r_{t-1}$.

**Q-Learning and Deep Q-Networks** The action-value function $Q^\pi$ gives the estimated return when the agent has the state history $h_t$, executes action $a_t$ and follows the policy $\pi$ on the future time steps. It can be recursively defined by $Q^\pi(h_t, a_t) = \mathbb{E}_{s_{t+1}, r_t \sim p}\left[r_t + \gamma \mathbb{E}_{a_{t+1} \sim \pi}[Q^\pi(h_{t+1}, a_{t+1})]\right]$. Q-learning and Deep Q-Networks (DQN) [25] are popular methods for obtaining the optimal action value function $Q^*$. Once we have $Q^*$, the optimal policy is also available as $\pi^* = \arg\max_{a_t} Q^*(h_t, a_t)$. In DQN, the action-value function is approximated by a deep NN with parameters $\theta$. $Q_\theta^*$ is found by minimising the loss function:

$$\mathcal{L}_t(\theta) = \mathbb{E}_{h_t, a_t, r_t, h_{t+1}}[(y_t - Q_\theta^\pi(h_t, a_t))^2], \quad \text{where } y_t = r_t + \gamma \max_{a'} Q_{\theta'}^\pi(a_{t+1}, h_{t+1}), \quad (2)$$

where $\pi$ is the $\epsilon$-greedy policy which takes action $\arg\max_{a_t} Q^\pi(a_t, h_t)$ with probability $1 - \epsilon$, and takes a random action with probability $\epsilon$. $\theta'$ are the parameters of a target network that are periodically copied from $\theta$ and kept constant for a number of iterations.

**Multi-Agent Reinforcement Learning** In this work, we consider the MARL setting where the underlying environment is modelled by a partially observable stochastic game [13]. In this setting, the environment is populated by multiple agents which have individual observations and rewards and act according to individual policies. Their goal is to maximise their own expected return.

# 4 Evolution via Evolutionary Reward

In this section, we propose a reward function that enables RL algorithms to search for policies with increasingly evolutionary success. We call this reward the evolutionary reward because it is always aligned with the fitness function. We also propose a specific RL algorithm that is particularly suited to maximise the evolutionary reward in open-ended evolutionary environments however other RL algorithms could also be used.

**Evolutionary reward** The evolutionary reward of an agent is proportional to the number of copies its genes have in the world's population. Maximising this reward leads to the maximisation of the survival and reproduction success of the genes an agent carries. We start by defining the kinship function between a pair of agents $i$ and $j$, who carry $N$ genes represented by the integer vectors $\boldsymbol{g}^i$ and $\boldsymbol{g}^j$ (we chose to use $\boldsymbol{g}$ for genome, which in biology is the set of genes an agent carries):

$$k \colon \mathbb{Z}^N \times \mathbb{Z}^N \to [0, 1], \qquad k(\boldsymbol{g}^i, \boldsymbol{g}^j) = \frac{1}{N} \sum_{p=1}^{N} \delta_{g_p^i, g_p^j} \quad, \tag{3}$$

where $\delta_{g_p^i, g_p^j}$ is the Kronecker delta which is one if $g_p^i = g_p^j$ and zero otherwise. When agent $i$ is alive at time $t + 1$, it receives the reward:

$$r_t^i = \sum_{j \in \mathcal{A}_{t+1}} k(\boldsymbol{g}^i, \boldsymbol{g}^j), \tag{4}$$

where $\mathcal{A}_{t+1}$ is the set of agents alive at the instant $t + 1$. Note that since agent $i$ is alive at $t + 1$, $\mathcal{A}_{t+1}$ includes agent $i$. $T^i - 1$ is the last time step that agent $i$ is alive and so, at this instant, the agent receives its final reward which is proportional to the discounted sum of the number of times its genes will be present on other agents after its death:

$$r_{T^i - 1}^i = \sum_{t=T^i}^{\infty} \gamma^{t - T^i} \sum_{j \in \mathcal{A}_t} k(\boldsymbol{g}^i, \boldsymbol{g}^j), \tag{5}$$

with this reward function, the agents are incentivised to maximise the survival and replication success of the genes they carry. In the agent-centered view, the agents are incentivised to survive and replicate, but also to help their family (kin) survive and replicate; and to make sure that when they die their family is in a good position to carry on surviving and replicating. The degree of collaboration with other family members depends on the overlap between their genotype as it happens in nature.

The discount factor, $\gamma$, needs to be in the interval $[0, 1[$ to ensure the final reward remains bounded. Due to the exponential discounting we can compute the final reward up to an error of $\epsilon$ by summing over a finite period of time denoted by the effective horizon ($h_e$). To see how to compute the $h_e$ for a given environment and $\epsilon$ see the Appendix G.1. By computing the final reward this way, we can now use RL algorithms like Q-learning to train agents with this evolutionary reward. However, in the next section we introduce a more practical algorithm that allows us to estimate the final reward more efficiently.

**Evolutionary Value-Decomposition Networks**   We propose Evolutionary Value-Decomposition Networks (E-VDN) as an extension of VDN [38] (explained in the Appendix D) from the binary cooperative setting with static teams to the continuous cooperative setting with dynamic families. E-VDN helps us reduce the variance of the value estimation and allows us to estimate the final evolutionary reward without having to simulate the environment forward for $h_e$ iterations.

Within a team, each agent fully cooperates with all the other members of the team, and it does not cooperate at all with any agent outside of the team. Moreover, if $a$ and $b$ are members of the same team and $c$ is a member of $a$'s team then $c$ and $b$ are also in the same team. Within a family, the degrees of cooperation amongst its members depends on their kinship degree (which can be any real number from 0 to 1). Also, if $a$ and $b$ are members of the same family and $c$ is part of $a$'s family, $c$ is not necessarily part of $b$'s family.

Each agent $i$ sees the members of its family from an unique perspective, based on the kinship degree it shares with them. In E-VDN, each agent $i$ has a joint action-value function, $Q^i$. E-VDN assumes $Q^i$ can be composed by averaging the action-value functions across the members of $i$'s family weighted by their kinship with agent $i$ (this is similar to the VDN's assumption):

$$Q^i((h_t^1, h_t^2, \ldots, h_t^{|\mathcal{A}_t|}), (a_t^1, a_t^2, \ldots, a_t^{|\mathcal{A}_t|})) \approx \frac{1}{n_t^i} \sum_{j \in \mathcal{A}_t} k(\boldsymbol{g}^i, \boldsymbol{g}^j) \tilde{Q}^j(h_t^j, a_t^j | \tilde{\theta}_j), \qquad (6)$$

where $n_t^i$ is a normalisation coefficient defined as $n_t^i = \sum_{j \in \mathcal{A}_t} k(\boldsymbol{g}^i, \boldsymbol{g}^j)$, $\tilde{Q}_t^j$ is the output of a NN with parameters $\tilde{\theta}_j$ and with the input $(h_t^j, a_t^j)$. Composing $Q^i$ with an average, instead of a sum as it happens in VDN, is necessary as E-VDN allows the number of value functions contributing to the composition to vary as the family gets bigger or smaller (agents born and die). This averaging allows us to incorporate the local observations of each family member and reduce variance in the value estimation.

More importantly, E-VDN allows us to deal with the difficulty of estimating the final reward (5) in a particularly convenient way. As is clear from its definition (5), the final reward is the expected sum (over time) of kinship that agent $i$ has with other agents $j$ after its death. The key idea is to note that this value ($r_{T^i-1}^i$) can be approximated by the Q-value of other agents $j$ that are close to (have high kinship with) agent $i$:

$$\hat{r}_{T^i-1}^i = \begin{cases} \frac{1}{n_{T^i}^i} \sum_{j \in \mathcal{A}_{T^i}} k(\boldsymbol{g}^i, \boldsymbol{g}^j) \tilde{Q}_{T^i}^j(\ldots) \approx Q_{T^i}^i(\ldots) & \text{if } n_{T^i}^i > 0 \\ 0 & \text{if } n_{T^i}^i = 0 \end{cases} \qquad (7)$$

The final reward is zero if, and only if, at the time of its death the agent has no surviving family.

Each $\tilde{Q}_t^i$ is trained by back-propagating gradients, $g_t^i$, from the Q-learning rule:

$$g_t^i = \nabla \boldsymbol{\theta_i} (y_t^i - \frac{1}{n_t^i} \sum_{j \in \mathcal{A}_t} k(\boldsymbol{g}^i, \boldsymbol{g}^j) \tilde{Q}^j(h_t^j, a_t^j | \tilde{\theta}_j))^2 \approx \nabla \boldsymbol{\theta_i} (y_t^i - Q_t^i(\ldots | \boldsymbol{\theta_i}))^2, \qquad (8)$$

where $\boldsymbol{\theta}_i$ is the concatenation of all the parameters $\tilde{\theta}_j$, used in each $\tilde{Q}^j$, contributing to the estimation of $Q^i$; i.e. $\boldsymbol{\theta}_i := \{\tilde{\theta}_j\}_{j \text{ s.t. } k(\boldsymbol{g}^i, \boldsymbol{g}^j) > 0}$. Note that $\tilde{Q}^i$ are neural networks with parameters $\tilde{\theta}_i$ and $Q^i$ is simply the average stated in (6).

The learning targets $y_t^i$ are given by:

$$y_t^i = \begin{cases} r_t^i + \gamma \max_{\boldsymbol{a}_{t+1}} Q_{t+1}^i(\ldots) | \boldsymbol{\theta'_i}) & \text{if } t < T^i - 1 \\ \hat{r}_{T^i-1}^i & \text{if } t = T^i - 1 \end{cases}, \qquad (9)$$

$r_t^i$ is the evolutionary reward (4), $\hat{r}_{T^i-1}^i$ is the estimate of the final evolutionary reward (7) and $\boldsymbol{\theta'_i}$ are the parameters of the target network that get periodically copied from $\boldsymbol{\theta_i}$. We don't use a replay

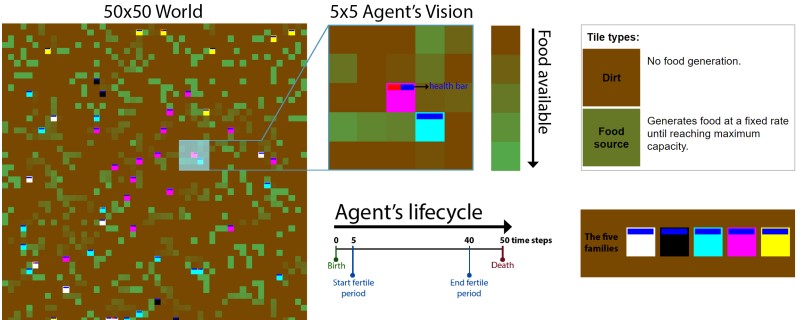

Figure 1: The binary environment.

buffer in our training (which is commonly used in DQN) due to the non-stationary of multi-agent environments (more about this in the Appendix G.2).

Since the joint action-value $Q^i$ increases monotonically with increasing $\tilde{Q}^i$, an agent acting greedily with respect to its action-value function will also act greedily in respect to its family action-value function: $\arg\max_{a_t^i} Q_t^i(\dots) \approx \arg\max_{a_t^i} \tilde{Q}^i(h_t^i, a_t^i)$.

## 5 Experimental Setup

We want to test two hypotheses: 1) E-VDN is particularly well suited to make agents climb the fitness landscape in open-ended evolutionary environments; 2) E-VDN is able to increase the evolutionary fitness of agents in non-binary cooperative environments. To test the first hypothesis we need to compare E-VDN with another popular evolutionary algorithm. To make it easier to implement the competing algorithm we are going to use a binary cooperative environment to test the first hypothesis. To test the second hypothesis we will use a non-binary cooperative environment. Note, if an agent carries more than one gene (like it happens in nature) we have a non-binary environment.

In this section, we give a quick overview of these two multi-agent environments, as well as details of the network architectures and the training regime. For a more complete description of the environments, you can refer to the Appendix E. In the binary environment, we compared our algorithm with a popular Evolution Strategies algorithm (CMA-ES [14]), and describe the training regime used for CMA-ES in the Appendix F.

**The Binary Environment** The binary environment is a 2-dimensional grid world, which is initialised with five agents carrying five unique genomes (Figure 1). At each time step, each agent may move one step and produce an attack to another agent in an adjacent tile. When an agent moves to a tile with food it collects all the food available in it. If an agent chooses to produce an attack, it decreases its victim's health by one point, if the victim's health reaches zero it dies and 50% of its collected food is captured by the attacker. The food is used to survive (one unit of food must be consumed every time step to remain alive), and to reproduce. When agents are within their fertile age and they have stored enough food, they reproduce themselves asexually and give birth to an agent carrying an exact copy of their genome. Each genome has only a single gene and there are no mutations. These rules make the cooperation between agents binary, agents either fully-cooperate (they have the exact same genome) or they don't cooperate at all (their genome has no overlap).

**The Non-binary Environment** The non-binary environment has the same rules as the binary environment with the difference that the agents now have 32 genes in their genome and they reproduce sexually instead of asexually. When two fertile agents are adjacent, they give birth to an agent who's genome is composed by two halves of the genes of each parent, selected randomly. There are no genders, any agent can reproduce with any other agent. These rules give rise to different levels of collaboration: from 0 to 1 in steps of $\frac{1}{32}$.

**Policy** Each agent observes a 5x5 square crop of the surrounding state (Figure 1). The agent sees six features for every visible tile; i.e. the input is a 5x5x6 tensor. This includes two features

corresponding to tile properties (food available and whether it is occupied or not) and four features corresponding to the occupying agents' properties (age, food stored, kinship and health). Besides these local inputs, each agent also observes its absolute position, family size and the total number of agents in the world. We intend to remove these extra inputs in future work as we provide agents with memory (we're currently providing our policy with $o_t^i$ instead of $h_t^i$). The NN has ten outputs (five movement actions with no attack and five movement actions with an attack). In this work, we used two different feed forward architectures: one is simply a fully connected NN with three hidden layers and $244,288$ parameters in total, the other architecture is composed by convolutional and dense layers and it is much smaller containing only $23,616$ parameters. The smaller NN was used to compare our algorithm with an evolutionary algorithm which doesn't scale well to larger networks.

**Training details**    In this work, the genome does not directly encode the policy, however, we think it would be interesting to do that in future work. In the binary environment, we train five different policies (with the same architecture but different weights) simultaneously. At each training episode, we sample five policies with replacement and assign each one to one of the five unique genomes. We do this, to force each policy to interact with all other policies (including itself), increasing their robustness in survival and reproduction. During the test episodes, no sampling occurs, each policy is simply assigned to each unique genome. The training episodes had a length between 450 and 550 (note that the reward is computed as if there was no episode end), and the test episodes had a length of 500 steps.

In the non-binary environment, due to the large number of unique genomes, it is unfeasible to assign a unique policy to each unique genome. To keep things simple, we chose to use only one policy in this environment. This was not possible to do with CMA-ES, so we did not implement it in this environment (more about CMA-ES on Appendix F).

**Traits encoded by the genes**    In the non-binary environment, we can think of each of the 32 genes to change some visual feature (e.g. facial feature) of their agent so that it can be better recognised by its family. In the binary environment, besides the gene encoding this visual feature it also encodes which policy, chosen from a set of 5 policies, the agent is going to have. Note that the genes encode fixed traits (they don't change during an agent's lifetime) and their frequency in the population evolve through normal evolution (death and birth). With EvER we don't need evolution to create the reward function and continuously align it with the fitness function. The agent's brain is always trying to learn the right things for the survival of its genes, however, the actual genes are evolving at the normal pace of evolution.

To analyse the impact of our reward function, we deliberately chose to minimise entanglement between genes and other aspects of the agents. However, EvER can be easily used in environments where genes encode more traits like the agent's abilities, visual features, initial weights and the topology of its policy.

**Evaluation Metrics**    In our simple environments, fitter policies can use the environment resources more efficiently and increase their population size to larger numbers. Therefore, to evaluate the performance of the algorithms in generating increasingly fitter species we track the average population size along training time.

# 6    Results

Training agents with E-VDN generates quite an interesting evolutionary history. Throughout the binary environment history, we found four distinct eras where agents engage in significantly distinct behaviour patterns (1$^{st}$ row of fig. 2). In the first era (the blue line - which lasts only a few hundred iterations), the agents learned how to survive, and through their encounters with the other founding agents, they have learnt that it was always (evolutionary) advantageous to attack other agents. In the second era (orange line), the agents' food-gathering skills increased to a point where they started to reproduce. In this era, the birth-rate and population numbers increased fast. However, with the extra births, intra-family encounters became more frequent, and intra-family violence rose to its all-time maximum driving the average life span down. This intra-family violence quickly decreased in the third era (green line), as agents started to recognize their kin. Kin detection allowed for selective kindness and selective violence, which took the average life span to its all-time maximum. Finally,

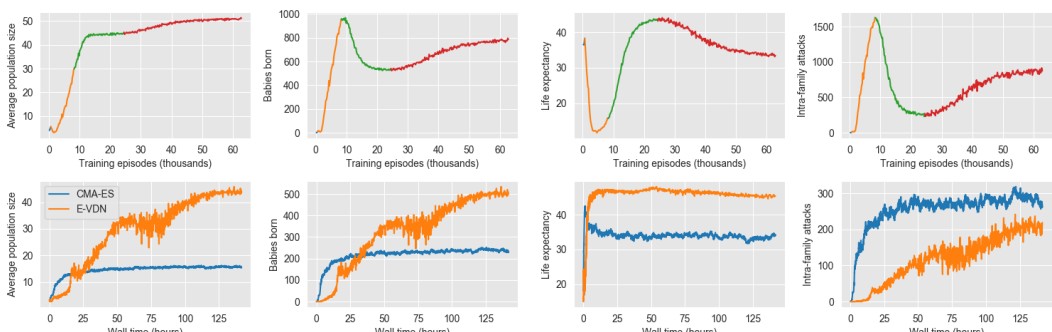

Figure 2: (1st row) Results obtained using E-VDN with the larger NN, each point was obtained by averaging 20 test episodes. The different colours correspond to different eras. This plot was generated with a denser version of the evolutionary reward (more details on the Appendix G.3). (2nd row) Results obtained using CMA-ES and E-VDN algorithms with the smaller NN and the standard evolutionary reward (4). Both algorithms were trained with 20 CPUs each.

in the fourth era (red line), agents learned how to sacrifice their lives for the future of their family. Old infertile agents started allowing the younger generation to eat them without retaliation. Through this cannibalism, the families had found a system for wealth inheritance. A smart allocation of the family's food resources in the fitter generation led to an increase in the population size with the cost of a shorter life span. This behaviour emerges because the final reward (5) incentivises agents to plan for the success of their genes even after their death. This behaviour is further investigated in the Appendix H.1. These results show that optimising open-ended evolutionary environments with E-VDN does indeed generate increasingly complex behaviours.

The 2nd row of Figure 2, shows the macro-statistics obtained by training the smaller NN with CMA-ES and E-VDN. From the figure, we observe that E-VDN is able to produce a larger population of agents with a longer life-span and a higher birth rate. A small population means that many resources are left unused by the current population, this creates an opportunity for a new and more efficient species to collect the unused resources and multiply its numbers. These opportunities are present in the CMA-ES environment, however the algorithm could not find them, which suggests that E-VDN is better at finding the way up the fitness landscape than CMA-ES. Video 1, shows that each family trained with CMA-ES creates a swarm formation in a line that moves around the world diagonally. When there is only one surviving family, this simple strategy allows agents to only step into tiles that have reached their maximum food capacity. However, this is far from an evolutionarily stable strategy [35] (ESS; i.e. a strategy that is not easily driven to extinction by a competing strategy), as we verify when we place the best two families trained with CMA-ES on the same environment as the best two E-VDN families and observe the CMA-ES families being consistently driven quickly to extinction by their competition (fig. 4.a of Appendix B).

Our results, in the non-binary environment, show that in a non-binary cooperative setting E-VDN also improves the ability of the trained policy to survive and replicate its genes (Figure 4.b,c and d of Appendix B). This is a key feature that evolutionary algorithms should have in order to take the research in open-ended evolutionary environments further. Note, that the non-binary environment is much harder than the binary one. To replicate, agents need to be adjacent to other agents. In the beginning, all agents are unrelated making it dangerous to get adjacent to another agent as it often leads into attacks, but it is also dangerous to get too far away from them since with a limited vision it is hard to find a fertile mate once they lose sight of each other. Video 2 shows a simulation of the evolved policy being run on the non-binary environment, it seems that agents found a way to find mates by moving to a certain region of the map (the breeding ground) once they are fertile.

## 7   Conclusion & Future Work

This paper has introduced an evolutionary reward function that when maximised also maximises the evolutionary fitness of the agent. This allows RL to be used as a tool for research of open-ended evolutionary systems. To implement this reward function, we extended the concept of team to the

concept of family and introduce continuous degrees of cooperation. Future work could explore three directions: 1) Explore a different reward function that makes agents maximise the expected geometric growth rate of their genes; 2) Research the minimum set of requirements to emerge natural cognitive abilities in artificial agents such as identity awareness and recognition, friendship and hierarchical status (by following our proposed methodology for progress in AI (Appendix **??**)) 3) Extend the use of genes to encode more fixed traits in the agent like its initial weights and the topology of its policy.

## Broader Impact

Simulating the key processes that generated life and intelligence in nature is a promising path to further our understanding in this field and unlock ever more intelligent algorithms able to solve useful problems for the world. However, embodying AI with the goal to survive and self-reproduce can be dangerous, and should never be done outside of a sand-boxed environment.

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
