## Appendix A    Videos

504   - Video 1 (`https://drive.google.com/open?id=1aVL7fmNp_rMpU2E0OPXq6BGtDNvmvCTu`)
505   - Video 2 (`https://drive.google.com/open?id=1s16Mz66ETV_dpLXi0yyaJLJHZpFN0luv`)

## Appendix B    Extra Figures

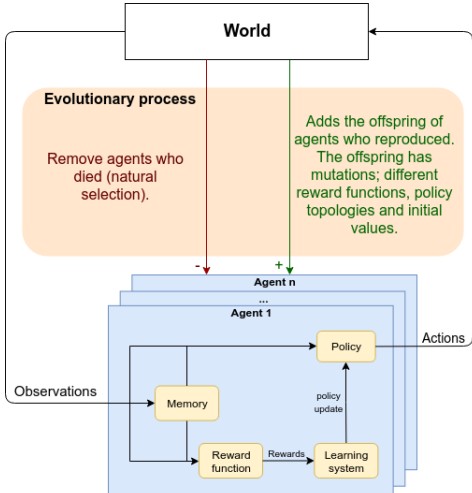

Figure 3: A diagram on how evolution and learning affect the development of the brain.

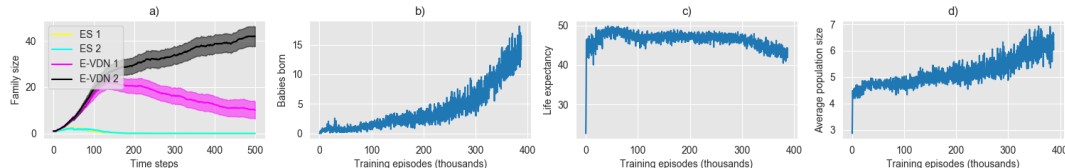

Figure 4: a) (CMA-ES vs E-VDN) Average and 95% confidence interval of two CMA-ES and two E-VDN family sizes computed over 90 episodes. b, c and d) the macro-statistics obtained in the non-binary environment. To speed up the training we used the smaller NN and the denser evolutionary reward described in the Appendix G.3.

## Appendix C    Optimally Aligned Reward: Formal Description

The notion of an optimal reward function for a given fitness function was introduced by Singh [21, 33], here we adapt his original formulation. From the perspective of agent $i$ the environment is defined by the state transition distribution; $E^i := p(s^i_{t+1}|s^i_t, a^i_t, \boldsymbol{\pi}^{-i})$. Where $\boldsymbol{\pi}^{-i}$ is the concatenation of the policies of all agents except agent $i$; $\boldsymbol{\pi}^{-i} := \{\pi^j\}_{\forall j \neq i}$. Formally we say: $h^i_\pi \sim \langle \mathcal{L}(\mathcal{R}^i), E^i \rangle$, where $h^i_\pi$ is the sampled history of the adaptations of policy $\pi^i$ which resulted from agent $i$ learning $\mathcal{L}(\cdot)$ to maximise its reward function $\mathcal{R}^i$ by interacting $\langle \cdot \rangle$ with the environment. If agent $i$ is the only agent learning then $\boldsymbol{\pi}^{-i}$ is static and so is the environment $E^i$. In this case, the optimally aligned reward function is given by:

$$\mathcal{R}^* = \arg\max_{\mathcal{R}^i} \mathbb{E}_{h^i_\pi \sim \langle \mathcal{L}(\mathcal{R}^i), E^i \rangle} \mathcal{F}(h^i_\pi, E^i), \tag{10}$$

where $\mathcal{F}$ is the fitness function. In the general case, all agents are learning, and therefore, the environment is non-static, the fitness for $h^i_\pi$ is changing and so is the optimally aligned reward $\mathcal{R}^*$.

## Appendix D  Value-Decomposition Networks

Our work builds on VDN [38], which was designed to address the binary cooperative MARL setting. In this setting, the agents are grouped into teams and all the agents within a team receive the same reward. VDN's main assumption is that the joint action-value function of the whole team of cooperative agents can be additively decomposed into the action-value functions across the members of the team.

$$Q^{\mathcal{T}}((h_t^1, h_t^2, \ldots, h_t^{|\mathcal{T}|}), (a_t^1, a_t^2, \ldots, a_t^{|\mathcal{T}|})) \approx \sum_{i \in \mathcal{T}} \tilde{Q}^i(h_t^i, a_t^i), \tag{11}$$

where $\mathcal{T}$ is the set of agents belonging to the team, and $\tilde{Q}^i(h_t^i, a_t^i)$ is the value function of agent $i$ which depends solely on its partial observation of the environment and its action at time $t$. $\tilde{Q}^i$ are trained by back-propagating gradients from the Q-learning rule through the summation.

$$g_i = \nabla \theta_i (y_t^{\mathcal{T}} - \sum_{i \in \mathcal{T}} \tilde{Q}(h_t^i, a_t^i | \theta_i))^2, \qquad y_t^{\mathcal{T}} = r_t^{\mathcal{T}} + \gamma \sum_{i \in \mathcal{T}} \max_{a_{t+1}^i} \tilde{Q}(h_{t+1}^i, a_{t+1}^i | \theta_i), \tag{12}$$

where $\theta_i$ are the parameters of $\tilde{Q}^i$, $g_i$ is its gradient and $r_t^{\mathcal{T}}$ is the reward for the team $\mathcal{T}$ at the time instant $t$. Note that even though the training process is centralised, the learned agents can be deployed independently, since each agent acting greedily with respect to its own $\tilde{Q}^i$ will also maximise its team value function $\arg\max_{a_t^i} Q_t^{\mathcal{T}}(\ldots) \approx \arg\max_{a_t^i} \tilde{Q}^i(h_t^i, a_t^i)$.

## Appendix E  Environment

In this section, we go through the game loop of the environments summarised in the main article. Both the binary and non-binary environment have the same game loop, their only difference is in the way agents reproduce and in the length of the genome agents carry (the genome has a single gene in the binary environment and 32 genes in the non-binary one). The states of the tiles and agents are described in table 1.

| | Tile state | Agent state | |
|---|---|---|---|
| Type | Boolean (food source/dirt) | Position (x,y) | Integer, Integer |
| Occupied | Boolean | Health | Integer |
| Food available | Float | Age | Integer |
| | | Food stored | Float |
| | | Genome | Integer Vector |

Table 1: The state of the tiles and agents.

We now introduce the various components of the game loop:

**Initialisation**  The simulation starts with five agents, each one with a unique genome. All agents start with age 0 and $e$ units of food (the endowment). The environments are never-ending. Table 2 describes the configuration used in the paper.

| Endowment $(e)$ | Initial health | Start of fertility age | End of fertility age | Longevity | World size | Food growth rate $(f_r)$ | Maximum food capacity $(c_f)$ |
|---|---|---|---|---|---|---|---|
| 10 | 2 | 5 | 40 | 50 | 50x50 | 0.15 | 3 |

Table 2: Configuration of the environment used in the paper.

**Food production**  Each tile on the grid world can either be a food source or dirt. Food sources generate $f_r$ units of food per iteration until reaching their maximum food capacity ($c_f$).

**Foraging** At each iteration, an agent can move one step to North, East, South, West or choose to remain still. When an agent moves to a tile with food it collects all the available food in it. The map boundaries are connected (e.g. an agent that moves over the top goes to the bottom part of the map). Invalid actions, like moving to an already occupied tile, are ignored.

**Attacking** At each iteration, an agent can also decide to attack a random adjacent agent: this is an agent within one step to N, E, S or W. Each attack takes 1 unit of health from the victim's. If the victim's health reaches zero, it dies, and the attacker will "eat it" and receive 50% of its food reserves.

**Asexual Reproduction** An agent is considered fertile if it has accumulated more than twice the amount of food it received at birth (i.e. twice its endowment $e$) and its age is within a given fertile age. The fertile agent will give birth once they have an empty tile nearby, when that happens the parent transfers $e$ units of food to its newborn child. The newborn child will have the same genome has its parent.

**Sexual Reproduction** An agent is considered fertile if it has accumulated more than the amount of food it received at birth and its age is within a given fertile age. The fertile agent will give birth once it is adjacent to another fertile agent and one of them has an empty tile nearby, when that happens each parent transfers $\frac{e}{2}$ units of food to its newborn child. A random half of the newborn's genes come from the first parent, and the second half comes from the second parent.

**Game loop** At every iteration, we randomise the order at which the agents execute their actions. Only after all the agents are in their new positions, the attacks are executed (with the same order as the movement actions). The complete game loop is summarized in the next paragraph.

At each iteration, each agent does the following:

- Execute a movement action: Stay still or move one step North, East, South or West.

- Harvest: Collect all the food contained in its tile.

- Reproduce: Give birth to a child using asexual or sexual reproduction (see their respective sections).

- Eat: Consume a unit of food.

- Age: Get one year older.

- Die: If an agent's food reserves become empty or it becomes older than its longevity threshold, then it dies.

- Execute an attack action: After every agent has moved, harvested, reproduced, eaten and aged the attacks are executed. Agents that reach zero health get eaten at this stage.

Additionally, at each iteration, each food source generates $f_r$ units of food until reaching the given maximum capacity ($c_f$).

## Appendix F  Evolution Strategies

In the binary environment, we compare the E-VDN algorithm with a popular ES algorithm. ES algorithms optimise an agent's policy by sampling policy weights from a multivariate Gaussian distribution, evaluating those weights on the environment, giving them a fitness score and updating the Gaussian distribution parameters so that the next samples are more likely to achieve a higher fitness. There are a few different methods on how to update the distribution parameters, we chose to use CMA-ES [14] because it has been successful in optimising NN for a wide range of sequential decision problems [16, 17, 18]. However, note that CMA-ES was not designed for multi-agent settings where the fitness function landscape changes as the other agents learn. Nevertheless, we used five independent multivariate Gaussians distributions each one associated with a unique gene and each one being updated by the CMA-ES algorithm. In the beginning, when the agents can not survive for long, the fitness function is given by the total sum of family members along time, once the agents learn how to survive and reproduce we change the fitness function to be the number of family members at the end of an episode with 500 steps. Since the CMA-ES algorithm computation

time grows quadratic with the number of parameters, $O(N^2)$, we had to use the smaller NN for this comparison. The algorithm was implemented using an available *python* library [15].

In the non-binary environment, if the initial five agents have different policies it creates the problem of deciding which policy should a child inherit. At the same time, the initial five agents can't all share the same policy because it then becomes impossible to define the fitness function for each policy. For these reasons, we didn't implement CMA-ES on the non-binary environment.

# Appendix G    Algorithm details

## G.1    Effective time horizon

We want to find the number of iterations ($h_e$) that guarantee an error between the estimate of the final reward and the actual final reward to be less or equal than a given $\epsilon$, $|r_{T^i-1}^i - \hat{r}_{T^i-1}^i| \leq \epsilon$.
Remember that the final reward is given by:

$$r_{T^i-1}^i = \sum_{t=T^i}^{\infty} \gamma^{t-T^i} \sum_{j \in \mathcal{A}_t} k(\boldsymbol{g}^i, \boldsymbol{g}^j) = \sum_{t'=0}^{\infty} \gamma^{t'} k_{t'}^i$$

Where $t' = t - T^i$ and $k_{t'}^i = \sum_{j \in \mathcal{A}_{t'}} k(\boldsymbol{g}^i, \boldsymbol{g}^j)$. The estimate of the final reward is computed with the following finite sum $\hat{r}_t^i = \sum_{t'=0}^{h_e-1} \gamma^{t'} k_{t'}^i$.

Note that $k_t^i$ is always positive so the error $r_{T^i-1}^i - \hat{r}_{T^i-1}^i$ is always positive as well. To find the $h_e$ that guarantees an error smaller or equal to epsilon we define $r_b$ as the upper bound of $k_t^i$ and ensure that the worst possible error is smaller or equal to epsilon:

$$\sum_{t'=0}^{\infty} \gamma^{t'} r_b - \sum_{t'=0}^{h_e-1} \gamma^{t'} r_b \leq \epsilon \tag{13}$$

$$\frac{r_b}{1-\gamma} - r_b \frac{1-\gamma^{h_e}}{1-\gamma} \leq \epsilon \tag{14}$$

$$\frac{r_b \gamma^{h_e}}{1-\gamma} \leq \epsilon \tag{15}$$

$$h_e \log \gamma \leq \log \frac{\epsilon(1-\gamma)}{r_b} \tag{16}$$

$$h_e \leq \frac{\log \frac{\epsilon(1-\gamma)}{r_b}}{\log \gamma} \tag{17}$$

We go from (1) to (2) by using the known convergences of geometric series: $\sum_{k=0}^{\infty} ar^k = \frac{a}{1-r}$ and $\sum_{k=0}^{n-1} ar^k = a\frac{1-r^n}{1-r}$ for $r < 1$. Since $h_e$ needs to be a positive integer we take the ceil $h_e = \left\lceil \frac{\log \frac{\epsilon(1-\gamma)}{r_b}}{\log \gamma} \right\rceil$ and note that this equation is only valid when $\frac{\epsilon(1-\gamma)}{r_b} < 1$. For example, an environment that has the capacity to feed at most 100 agents has an $r_b = 100$ (which is the best possible reward, i.e. the kinship between every agent is 1). If we use $\epsilon = 0.1$ and $\gamma = 0.9$ then $h_e = 88$.

## G.2    Experience buffer

When using Q-learning methods with DQN, as we are, it's common practice to use a replay buffer. The replay buffer stores the experiences $(s_t, a_t, r_t, s_{t+1})$ for multiple time steps $t$. When training, the algorithm randomly samples experiences from the replay buffer. This breaks the auto-correlation between the consecutive examples and makes the algorithm more stable and sample efficient. However, for non-stationary environments, past experiences might be outdated. For this reason, we don't use a replay buffer. Instead, we break the auto-correlations by collecting experiences from many independent environments being sampled in parallel. After a batch of experiences is used we discard

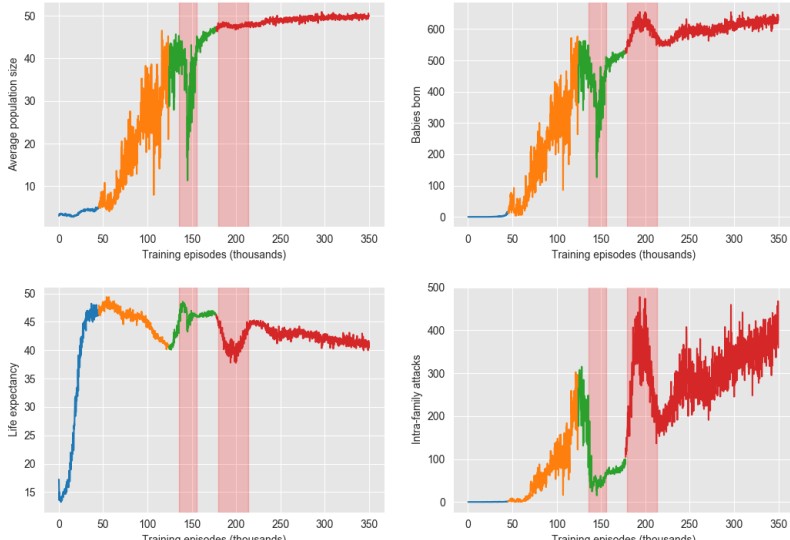

Figure 5: Macro-statistics when evolving bacteria using the standard evolutionary reward. I: learning to survive (blue line), II: learning to reproduce (orange), III: learning to detect kin (green), IV: learning to self-sacrifice (red). The red bands correspond to the First and Second Family Wars.

them. In our experiments, we simulated 400 environments in parallel and collected one experience step from each agent at each environment to form a training batch.

### G.3 Denser reward function

In some situations, we used a denser version of the evolutionary reward to speed up the training process. We call it the *sugary* reward, $r_t'^i = \sum_{j \in \mathcal{A}_t} k(\boldsymbol{g}^i, \boldsymbol{g}^j) f_t^j$ where $f_t^j$ is the food collected by agent $j$ at the time instant $t$. In these simple environments, the *sugary* and the evolutionary reward are almost equivalent since a family with more members will be able to collect more food and vice-versa. However, the *sugary* reward contains more immediate information whilst the evolutionary reward has a lag between good (bad) actions and high (low) rewards; a family that is not doing a good job at collecting food will take a while to see some of its members die from starvation. Nonetheless, the evolutionary reward is more correct since it describes exactly what we want to maximise. Note that this reward was not used to produce the results when comparing E-VDN with CMA-ES.

When using the standard evolutionary reward to evolve the larger NNs, the same four eras, that were observed with the *sugary* reward, emerge. However, their progression is not as linear. In this case, the families take longer to learn and sometimes one family evolves much faster than the others. When this happens, the families left behind eventually catch up with the most developed ones. The behaviour of the emerging families successfully interferes with the developed ones creating a temporary disruption in the environment which disrupts its macro-statistics. Two disruptions were observed in one of our simulations and we named them the First and the Second Family Wars (fig. 5).

## Appendix H    Results details

### H.1    Cannibalism and suicide as a tool for gene survival

In the evolutionary history of the binary environment we saw the rise of cannibalism in the fourth era. Figure 6.a shows how the average age of cannibals and their victims grows apart in this era. After observing this behaviour, we wanted to know how important cannibalism was for gene survival. To answer this question, we measured the family size of a certain family when its members were not allowed to attack each other and compared it with the normal situation where intra-family attacks were allowed (see Figure 6.b). Figure 6.b clearly shows that, in this environment, cannibalism is essential for long-term gene survival. We also ran this exact experiment before the fourth era and

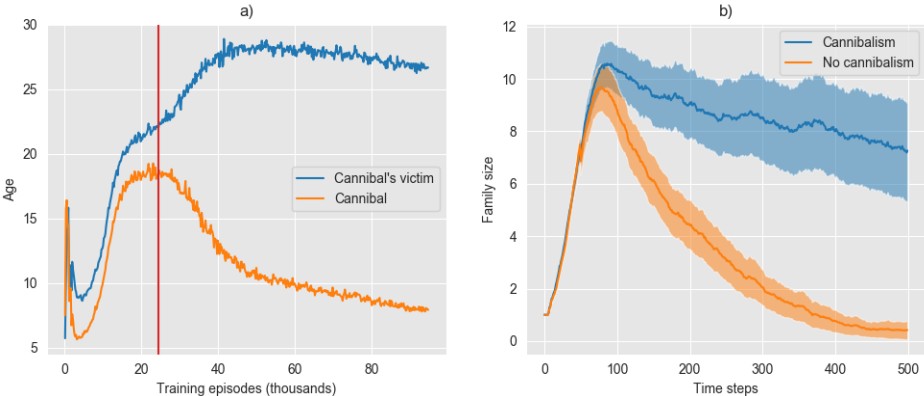

Figure 6: a) The average age of intra-family cannibals and cannibals' victims. The vertical red line marks the start of era IV. b) the size of family 1 averaged along 90 test episodes. To compute the orange line we simply blocked all the attacks between members of the family 1. The shaded bands represent the 95% confidence interval.

achieved the opposite results, suggesting that before this era the agents didn't yet know how to use cannibalism to their gene's advantage (results not shown).

## H.2 Genetic drift

The environment starts with a unique set of alleles[1], and there is no mechanism to add diversity to this initial set (no mutations) but there is a mechanism to remove diversity: death. When a death occurs, there is a chance that the last carrier of a particular allele is lost and if there are no mutations, this is a permanent loss in diversity. This evolutionary mechanism that changes allele frequencies using chance events is called genetic drift. In the binary environment, this means that if we simulate the environment for long enough all the agents will end up sharing the same allele. In the non-binary environment, all the agents will have the same genome, however, this genome will likely be composed by alleles coming from all the five founders (see Video 2).

We found that in the binary environment kin detection speeds up this decrease in allele diversity. This was expected since agents cooperate with kin and compete with non-kin. Therefore, as a family gets bigger, its members become more likely to encounter cooperative family members rather than competitive unrelated agents. This improves the survival and reproduction success of that family, making it even bigger. Figure 7 shows the decrease in diversity with and without kin selection (we removed kin selection by zeroing out the kinship feature in the agents' observations). From the figure, it is evident that kin selection speeds up the decrease in diversity. However, note that before the 100[th] iteration, kin detection leads to a slightly higher diversity. This happens because kin detection reduces intra-family violence, leading to fewer deaths and consequently to a slower genetic drift. The environment usually reaches its maximum capacity around the 100[th] iteration, at this time the inter-family competition is at its highest and the positive feedback loop created by kin selection starts having a larger importance.