# OpenReview forum: "Mimicking Evolution with Reinforcement Learning"
_NeurIPS.cc/2021/Conference — NeurIPS 2021 Submitted_

### Official Review · Reviewer_KvmB · 2021-07-11

**Rating:** 5
**Confidence:** 4

**Summary:**

** Score revised after reading through initial author comments **

The papers compares and contrasts evolutionary processes with learning processes and discusses some of the links between the two in reinforcement learning problems. Subsequently, the authors describe some previous work on combining evolution with learning processes and set up a reward function that encourages the maintenance propagation of agents in the kin based on genetic encodings. The authors hypothesize that this reward function is provides a useful learning signal in open-ended environments. According to their review of previous work, this reward function solves the alignment problem between learning and evolution by ensuring that the learning policy learns to take actions that encourage the survival of a set of agents of similar kin. Next, the authors formulate a method to optimize that particular reward function named E-VDN, which is an extension of value decomposition networks with E-VDn taking into account the gene marker make-up of a particular agent. The authors then show experiments in a set of multi-agent reinforcement learning environments and analyze the resulting behavior of their trained agents and compare the results of their methods to agents trained with CMA-ES.

**Limitations And Societal Impact:**

The discussion on limitations and societal impacts is pretty limited. Given that the current version still has almost an entire page left, I think it would be good to use the remaining space to address limitations and provide a more thorough discussion on potential impacts. Some things to think about could be:
* What kind of problems can this formulation solve? What kinds can it not solve?
* What could some potential drawbacks and mitigations to having more and more intelligent agents interacting with each other and humans in the real world?
* How expensive is it to train the current framework and how would that scale to larger, more complicated problems?

**Main Review:**

Nits:
* Line 114 and Line 350 have Appendix ?? in the text

I think that the paper introduces an interesting notion that one can encourage evolution like behavior in reinforcement learning by formulating a reward and framework that encourages the propagation of one's genetic code. In its current form, however, I think that paper still has a couple of major weaknesses relating to the clarity of some of the narrative, a limited set of experiments in only a set of particular environments with comparison to only one other method.

**Originality:** I think that the authors' idea of introducing a novel reward formulation and method to optimize that particular reward with a prescribed encoding of genetic markers into the method is a promising idea that could lead to interesting research directions.

**Quality:** I think that paper in its current form provides a lot of detail in the description of the method and the thought process behind it, but it does fall short on description of some of the RL specific settings pertaining to the experiments. I also think that paper could be significantly strengthened if additional experiments would be conducted to further tests some of the authors' hypothesis. Some specific suggestions:
* Provide a more thorough description of the MARL setting and its definition and it is particularly suited for your method. The MARL setting is different from the traditional RL setting given multiple agents, so it would be good to have more detail on that in the paper.
* Conduct more experiments in different environments that test your hypothesis further. The authors specifically mention OpenAI's Neural MMO, which would probably be a good choice for additional experiments. Further experiments comparing to other methods in MARL settings would further strengthen the paper, especially if those methods include evolution+RL algorithms, such as "Majumdar, Somdeb, et al. "Evolutionary Reinforcement Learning for Sample-Efficient Multiagent Coordination." International Conference on Machine Learning. PMLR, 2020." which also claims to use evolution to coordinate in multi-agent settings.
* Provide a  discussion on the limitations of your work that can address some of the following questions: In what kind of open-ended problems do you think the proposed formulation holds promise and in which ones does it not? It appears that your formulation specifically targets survival and reproduction, which I would argue is not ideal in all open-ended RL settings.

**Clarity:** The technical portion of the paper is relatively clear. Yet, I think that the background section (Section 3) could be significantly improved by providing more clarity and descriptions of the MARL setting. It might also be helpful to have a diagram that compares and contrasts the authors' method to other method, which could be combined with Figure 3 in the appendix.

**Significance:** I think that scope of the authors claims could be substantially improved with additional experiments in other environments and in comparisons with other methods.

**Time Spent Reviewing:**

2

---

> ### Author Response · Authors · 2021-08-09
> **Our contribution is significant and our results provide necessary evidence to back it: "we can optimise a reward function that is always aligned with evolution, and therefore, we don’t need to go through the process of aligning a reward function"**
>
> We deeply appreciate the time you took to read, understand and write constructive feedback to improve our work.
>
> We will fix the problem with the appendix.
>
> ### Comments on the summary:
> We’re glad to see that our paper was well understood.
>
> ### Comments on quality:
> - It would be great if you could indicate what else you would like to see in relation to the MARL setting. For instance, we could go deeper into the competitive settings where a team collaborates to defeat another team (as happens in StarCraft, Dota 2 and Google’s Football environment). Or talk more about the specific challenges of MARL in relation to RL like non-stationarity environments, the multi-agent credit assignment problem, predictability vs exploitability or the lazy-agent problem.
> - It would be great to run our algorithm on Neural MMO. After reading our paper, the author of Neural MMO has publicly stated (on twitter, link not given for author anonymity) that he would like to implement a variant of EvER in a future version of NMMO. Implementing EvER on NMMO is not straightforward since NMMO is not truly an evolutionary environment: there is no reproduction on NMMO. We also agree that it would be interesting to compare our algorithm against a procedure where evolution is optimising the agent's reward function while learning is optimising their policy. We find it interesting, but we also think that our current experiments show sufficient evidence to support our core hypothesis: we can optimise a reward function that is always aligned with evolution, and therefore, we don’t need to go through the process of aligning a reward function. The advantage of starting with an aligned reward function is clear when comparing it to a procedure that needs to align the reward function. The cost of aligning the reward function will depend on the evolutionary algorithm being used, the environment and the agents. The bigger the alignment cost is, the bigger the advantage of our method. It would be interesting to learn more about that but we feel that it wouldn’t provide more evidence to back our hypothesis.
> - We agree. Our formulation specifically targets evolutionary environments (environments where survival and reproduction are key). We believe there will be a growing set of environments with these properties to simulate biological evolution, economies and various sociological aspects. In all of these environments, we have replicators (genes, business ideas and memes) competing for a common limited-resource to survive and reproduce. We will write this more clearly in the paper.
>
> ### Comments on clarity:
> We agree and will improve these points.
>
> ### Comments on significance:
> Our claim is that we can optimise a reward function that is always aligned with evolution, and therefore, we don’t need to go through the process of aligning a reward function. We believe that our results provide the necessary evidence to back this claim.
>
> We agree that it would be interesting to study how the cost of aligning a reward function varies depending on the evolutionary algorithm, environment and agents being used. However, running these experiments would not provide more evidence to our claim.
>
> ### Comments on limitations and societal impact:
> We agree and will improve these points (contingent on having enough space left on the paper).

---

> > ### Comment · Reviewer_KvmB · 2021-08-12
> > **Clarification on the Questions Raised**
> >
> > Thank you for your reply and for providing further clarifications for your work and your approach. I wanted to clarify some of the questions you brought up:
> > - In regards to the MARL settings, I think it would be good to describe the formulation and underlying problems that are closest to your setup and then show how you are different from them. I think that the competitive settings (StarCraft, Dota2) provide a good starting point for describing how different species would compete in teams for resources and a common goal. Once you have set that up, you can then follow up to show how your evolution setup is different and present an alignment problem.
> > - I am inclined to agree with you that provide supporting evidence for showing that your approach can optimize a reward function that is always aligned to evolution. What I think would make the paper stronger would be if you could do one, or more, of the following:
> >     - Describe why you think an approach that has different kind of optimization setting, such as the one I mentioned in my original review, where alignment of the reward for a MARL problem is "solved" using a different approach.
> >     - It appears to me that you could adopt your formulation to those approaches, so treating some of the other methods as comparables would strengthen your argument, especially if you can collect experiment data.
> >
> > Based on this discussion, I have revised my score to a 5 and hope we can continue to engage in productive conversation.

---

> > > ### Author Response · Authors · 2021-08-13
> > > **Clarifying the novelty of our method by comparing the evolutionary vs the team-cooperation MARL settings**
> > >
> > > Thank you again for your valuable feedback it is being really useful and it will improve the clarity of the paper.
> > >
> > > We will start by addressing your second point. We believe that going into detail about the differences and similarities of the paper you mentioned in your original comment (“Evolutionary Reinforcement Learning for Sample-Efficient Multiagent Coordination”) will be helpful to understand why our setting is different and why it needs new algorithms.
> > >
> > > In their paper, they are in a cooperative setting with a clear objective: improve the air traffic, win a football match, move all the robots to certain points, etc. Each agent in each team needs to have a reward that is aligned with the team’s goal. The obvious choice is to simply use the team’s goal (e.g. win the football match). However, this reward is sparse, and therefore, it would take many episodes until an agent gets enough reward signal and find a good policy. The authors of the paper, use evolution to create a denser reward function, for each agent in the team, that is aligned with the team’s reward. The denser reward is easier (and faster) to maximise without changing the result of that maximisation: having a policy that is able to win the football match. Minor detail: in the paper, they say that the agent’s reward may be misaligned with the team’s reward, but ideally the two should be aligned.
> > >
> > > In our paper, we have an evolutionary setting (we also call it a continuously cooperative setting). In this setting, we have agents that carry replicators. We made the analogy that agents were animals and replicators were genes, but it is more general than that. For example, agents can be companies that have many shareholders (replicators) that want to survive (not go bankrupt) and replicate (invest in (or create) more companies). Companies that have overlapping shareholders want to cooperate (proportionally to the degree of the overlap), while companies with no overlapping shareholders are competing for clients' money. There are many more examples of evolutionary settings, all we need is to have agents competing for a limited resource to survive and replicate the replicators they carry. Note, some Evolutionary Algorithms, despite their name, work very differently from natural evolution and are not suitable for this evolutionary setting.
> > >
> > > This setting brings new problems:
> > > 1) The concept of a team is not enough to describe the affiliations of the agents (agents may want to collaborate more with some agents than others). Our paper introduced the concept of family for this reason.
> > >
> > > 2) It’s not immediately clear what the goal of the agents is. Evolution acts on the replicators and not on the agents. It’s a process that adds variability to the replicators (with mutations) and filters them out with natural selection. The replicators don’t have a goal, but the fact is that replicators that are not good at surviving will get extinct. Replicators that are good at surviving, give their agents skills that help them survive and replicate (a healthy body/good shareholder meetings). So from the agent’s perspective, the goal of evolution can be seen as maximising the survival and replication of the replicators they carry. One could argue that the reward function we came up with is obvious, but it wasn’t obvious to us and, most importantly, it isn't obvious to figure out how to optimise it.
> > >
> > > 3) This leads to the final problem. It’s hard to optimise this reward function. For two reasons: 1) usually computing a reward does not involve computing an infinite sum, and 2) previous works in MARL did not address the case where agents have various levels of cooperation with other agents.
> > >
> > > The obvious way to tackle this setting is with a bi-level optimization procedure where evolution optimises the genes for population size and the inner learning algorithm adapts the policy for an objective set by the outer loop. We improve this approach by introducing the evolutionary reward and a way to optimise it.
> > >
> > > In regards, to your first suggestion. We agree and believe that this exercise of describing another MARL setting in more detail helps to understand better the novelty of our setting and approach.

---

### Official Review · Reviewer_J3e1 · 2021-07-14

**Rating:** 6
**Confidence:** 4

**Summary:**

The paper is concerned with optimisation in an open-ended ALife environment.
The authors propose to directly maximise the number of current and future genetically similar agents via an adaptation of Value Decomposition Networks, named Evolutionary Value-Decomposition Networks (E-VDN).
This is in contrast to applying a bi-level optimization procedure where evolution optimises the genes for population size and the inner learning algorithm adapts the policy for an objective set by the outer loop.

**Limitations And Societal Impact:**

Limitations and potential negative societal impact is adequately discussed.

**Main Review:**

# Originality and Significance

The idea of directly optimising the number of genetically similar agents directly as the objective of an agent is an interesting and novel approach.
This could be an approach to largely reduce computational cost of bi-level evolutionary systems.
The authors introduce a value estimation method that allows doing so cheaply called E-VDN.
In the context of the mentioned objective, E-VDN allows estimating the future reward of an agent based on the rewards of similar agents.
This paper should be interesting to the Meta Learning, Evolutionary, and ALife NeurIPS community.

# Great insights into dynamics of environment

In terms of their analysis, I particularly like the observed stages in the environment dynamics induced by their method.
The stages involve attacking other agents, learning to reproduce, recognising their kin in terms of the attacking behavior, and sacrificing their lives for the future of their genetically similar kinds.

# Environment may not be open-ended

The authors often argue that their setup is open-ended.
From their analysis, it seems that the generated complexity is not open-ended but converges.
This is common for papers that seek open endedness, but it should be further clarified that this goal is not reached.

# The CMA-ES baseline is not used to optimise the genes but the policies

This is confusing to the reader because previously the word 'evolution' has always been used in the context of only optimising genes.
In the case of their CMA-ES baseline, the same 'evolutionary reward' for the inner loop is used with CMA-ES to optimise the policies for one particular gene.
The authors should clearly distinguish the use of the word 'evolution' in the two different contexts.

**Time Spent Reviewing:**

6

---

> ### Author Response · Authors · 2021-08-09
> **We agree with this review and we will follow the suggestions to further improve the paper**
>
> We deeply appreciate the time you took to read, understand and write constructive feedback to improve our work.
>
> ### Comments on summary, originality, and insights:
> We’re glad to see that our paper and our contribution were perfectly understood.
>
> ### Comments on Environment may not be open-ended:
> We agree. There is an upper bound on the generated complexity limited by the expressivity of the environment space and the agents' neural networks. We meant that the environment is open-ended because there is not a specific goal (a behaviour that we are aiming to obtain). However, we agree that open-endedness should be defined by a boundless upper-limit on complexity and we did not achieve that goal. We will state this clearly in the paper.
>
> ### The CMA-ES baseline is not used to optimise the genes but the policies:
> We agree. Both E-VDN and CMA-ES are aligning the policies with evolutionary fitness so that the policies maximise the survival and replication of the agent’s genes. However, the evolution in the genes’ distribution occurs naturally through the birth and death of agents. We need to state this distinction more clearly in the paper.

---

> ### Comment · Reviewer_J3e1 · 2021-09-01
> **Paper improvements**
>
> I am still excited about the proposed idea / generally alternatives to bi-level optimization and would like to see more attempts and discussions in the community.
> On the other hand, I have to acknowledge that the current claims by the authors that EvER (1) would speed up the computation of bi-level evolutionary methods and that (2) aligning fitness and reward is hard [in their setting], is not supported by sufficient empirical evaluation.
> As suggested in my previous comment, the authors should introduce a new baseline that performs fitness and reward alignment (related to lines 86 to 93) and directly compare it with their method. Their current CMA-ES baseline does not have such a bi-level structure.
>
> The authors pointed out that EvER works only in the MARL setting and we don't have a suitable bi-level baseline to compare to.
> To resolve this issue, I would suggest to either move to the single agent setting (if possible) or create an appropriate baseline.

---

> > ### Author Response · Authors · 2021-09-03
> > **We would be criticised for adding such a baseline**
> >
> > 1) Our replicator evolution is by definition multi-agent (as is natural evolution).
> > 2) The double-loop is still not feasible for multi-agent systems. There are good theoretical reasons on why no one has yet made the double-loop work in MARL and why our approach is more tractable. We would be criticised by including a baseline that 1) we came up with and, 2) does not work.

---

### Official Review · Reviewer_2ceV · 2021-07-16

**Rating:** 3
**Confidence:** 4

**Summary:**

The article proposes an evolutionary simulation where agents use reinforcement learning to optimise fitness (in the sense of natural selection, number of genes in the population). Some of the ideas are interesting, but overall the contribution is limited, the applicability of the results is narrow, and the comparisons against evolutionary algorithms is not particularly fair. Fundamentally, it appears as though the authors have created a simulation where _greenbeard_ genes (see Dawkins 1976) are favoured. Basically, attacking individuals pays off if and only if the cost-benefit ratio exceeds their relatedness. This is Hamilton's rule, and a greenbeard makes it so that the relatedness is observable. This is at least the case of their "binary environment" and there are not enough details in the other environment to discern if this is also the case, but some language seems to suggest it is. This is not a novel insight in evolutionary biology, and seeing it work in reinforcement learning agents is not interesting when it is set as the _explicit reward_. Were this to emerge, then it would be much more interesting, particularly without observed relatedness, but also significantly more challenging.

**Limitations And Societal Impact:**

No, the limitations of the work should be more clearly stated.

**Main Review:**

## Originality

The article provides a very simple 2D environment where agents can move around, gather resources, possibly by attacking other agents, and reproduce. There are many other 2D environments with similar settings, and in the field of ALife that the authors mention, there are many explorations of the dynamics of these. The key difference here is that they are using RL agents, and that they are setting the reward signal as the evolutionary fitness of the individuals. This shows some interesting evolutionary dynamics with distinct phases, but the overall setting is very limited, and not clear what the actual advancement of the state of the art would be. All results are completely predicted by evolutionary theory, and the supposed improvement in computational complexity is simply due to setting the fitness as the reward.

## Quality

The work should be much better contextualised with the relevant literature in ALife and Evolutionary Biology. both of which have a long history of understanding cooperation.

## Clarity

The article would benefit greatly from a rewrite, as many claims and concepts spanning evolution and reinforcement learning are thrown together with little context or justification. For instance, halfway through the abstract, the authors claim that optimisation of a reward function to align with evolution is computationally expensive. It is unclear what literature the authors are referring to. There have been a series of articles on using, for instance, population dynamics and evolutionary theory as optimisers (e.g.
Omidshafiei et al. Neural Replicator Dynamics, arXiv:1906.00190). Another example: the authors claim that learning maximises pleasure and minimises pain... but this is unclear what it refers to. And clearly not true for most living organisms. Perhaps the authors are claiming this is true for all Chordata? Even so, it would be controversial to say any learning is about this.

The authors go to great lengths to talk about how alignment of reward to evolutionary fitness is hard, and the all one can hope to achieve is some partial alignment. However, the main contribution of the article is literally just computing the evolutionary fitness and turning that into the reward of the agents. At the end of section 1.1, the authors claim that one requires the use of simulation because the reward is extrinsic to the agent. It is unclear what the authors mean by this.

## Significance

The results are expected by previous theory, which limits the utility of the contribution. The computational improvement claimed requires changing the reward signal to match the desired outcome (evolutionary fitness), which negates the interestingness of the approach.

## Detailed comments

In Section 1.2, the authors should cite Hamilton's 1964 papers when referring to _kin selection_. Although Maynard-Smith popularised the term, the theory behind it was developed by Hamilton.

In _Combining evolution and learning_ the authors should probably cite Wang, et al 2018. Evolving intrinsic motivations for altruistic behavior, arXiv:1811.05931.

In lines 99-100, the authors claim that EAs must discard most of the state-action trajectories. But this is not intrinsically true. It might be that this has been a practical choice, particularly for simple iterated social dilemma scenarios, but definitely not a hard limitation.

In line 114, I'm not sure what the authors mean by _convergent evolution_ being key properties of life on Earth. The authors seem to suggest that convergent evolution is somehow synonymous with cooperation?

The intuition behind their reward function comes very late. Only in section 4, already in page 4, does it get introduced. This needs to be put front and centre, as this is the main contribution of the paper. Also, the reward function used is likely to lead to the "lazy agent" effect (see Sunehag 2017, and Rashid 2018, cited in the article!), or if not, it should be discussed why.

In line 168, the authors use the word _genome_ where they probably mean _genotype_.

In lines 192-197 the authors use the word _team_ but referring to them as _cliques_ would be more appropriate. Also, the distinction between teams and families really ought to come earlier and be clearly explained.

The caption of the figures should have more information. They are stubs at the moment.

In section 5, the details need to be more clearly explained. The authors say things like: "if an agent carries more than one gene [...] we have a non-binary environment." What does this mean? Surely the environment shouldn't be dependent on the agents in it. This would be better explained as two different experimental setups, rather than two different environments with some restrictions on the genetics of the agents that can participate in them.

There are not enough details to know, for instance, what the initial health of the agents is. Nor clear what the distribution of food in the environment is. I presume there must be substantially more than 1 unit per cell, since the agents require 1 unit per step just to survive. Also, it is not clear if the food replenishes, or at what rate. There are also not enough details on the payoffs of interactions.

The mechanics of movement and attacking are underexplained. The authors claim 5 actions that are pure movement (4 directions, + no-op) and also 5 actions with attack... But what does it mean to attack without moving? How would interactions get resolved if two agents attack simultaneously? Why would an agent not simply attack all the time it is not near a relative?

**Time Spent Reviewing:**

2

---

> ### Author Response · Authors · 2021-08-09
> **The review missed the point of our paper**
>
> We deeply appreciate the time you took to read, understand and write constructive feedback to improve our work.
>
> ### Comments on the summary:
> The summary misses the point of the paper. The reviewer says that if our reward had emerged from evolution then the paper would be more interesting. However, that approach is exactly what we wanted to improve upon. Using evolution to emerge a reward function that increases evolutionary fitness is time consuming and computationally expensive.
>
> Our reward function ensures that learning is always aligned with evolution. This means that when an agent maximises its reward, by learning, it also maximises its evolutionary fitness.
>
> Also, note that to evolve kin selection, kin recognition or kin inference is required.
>
> ### Comments on originality:
> We agree that the improvement in computational complexity is due to setting the fitness as the reward. However, how to do this is not obvious or simple. The choice of the reward function is not obvious or unique; there are many reward functions that are aligned with evolutionary fitness. We ran experiments with two reward functions (equation 4 and Appendix G.3), but there are others to be explored like maximising the expected growth rate of genes instead of maximising the expected number of genes. It’s also not simple to optimise these reward functions. The naive approach to obtain the terminal reward requires computing a potentially infinite series, which is not feasible. We introduced E-VDN so that agents can estimate this terminal reward in a computationally efficient way.
>
> Even though all results are predicted by evolutionary theory, we find it interesting, and non-obvious, that four distinct phases have emerged during the training of our agents.
>
> ### Comments on clarity:
> Aligning a reward function is always more expensive than starting with an aligned function.
> The alignment process is usually very expensive. To get a fitness measurement of the reward function it’s necessary to let the agent learn to maximise it for a sufficient time, and therefore, getting fitness measurements is expensive. The cost of optimizing the reward functions given various fitness measurements will vary depending on the evolutionary algorithm, environment and agents being used. However, the alignment process will never be free and so our method will always have some advantage.
>
> In many cases, our reward function can be further evolved to an optimally aligned reward function, that is shaped in a way that it is easier to learn without changing the resulting policy (a policy that maximises the agent’s future number of genes). However, even in this case, agents that start with our aligned reward function should evolve an optimal aligned reward function faster than agents that start with a random reward function.
>
> We agree that we shouldn’t use the words “pleasure” and “pain” which are concepts that are not well defined. We’ll change that sentence to “maximise the reward function”.
>
> Our reward is extrinsic to the agent because it contains information that is not provided by the agent’s state. For example, the terminal reward is related to the future survival and reproduction of the agent’s genes. The agent will not be alive to assess the survival of its genes, but in a simulation, we can look forward, assess this survival and then give the reward signal to the agent.
>
> ### Comments on significance:
> This misses the point of our paper. We never claimed to make a contribution to evolutionary biology. Our contribution is to achieve the results expected by previous theory, faster.
>
> ### Detailed comments:
> We will cite those papers.
>
> If EA were to use the state-action trajectories it would find it extremely hard to solve the credit assignment problem: how the actions along the trajectories should be credited for the final fitness. This is the main reason to use RL when dealing with state-action trajectories.
>
> Convergent evolution is when unrelated species evolve the same solutions by being exposed to similar problems. If we want AI agents to evolve similar solutions to the ones animals evolved, we need to expose them to similar problems that life found on Earth. To do that, the environment needs to mimic key properties of life on Earth.
>
> We can mention the “lazy agent” problem. It’s a problem that should be considered for any cooperative setting. It is not a problem when learning decentralised policies with a centralized critic as it was done in this paper.
>
> We will change to genotypes.
>
> Team is the usual name used in the literature. We describe what a family is in line 76 and we describe what a team is on line 118. We left the mathematical description of family to the methods section (section 4 - Evolution via Evolutionary Reward).
>
> In Multi-Agent RL, when looking at the environment from the perspective of any single agent, the environment is dependent on the other agents in it. For this reason, it is commonly said that multi-agent environments are non-stationary.
>
> Details about the environment are in Appendix E. The initial health and other parameters are shown in Table 2.

---

### Decision · Program_Chairs · 2021-09-27

**Decision:**

Reject

**Comment:**

Meta-review of "Mimicking Evolution with Reinforcement Learning"

In nature, and in computational evolution literature, evolution is usually the outer loop of optimization while RL is the inner loop. This bi-level optimization is computationally expensive. This work proposes a way to do both in tandem - the authors propose to directly maximise the number of current and future genetically similar agents. They test the concept on 2 bio-inspired open-ended grid world environments.

Reviews were mixed for this work. Summary of discussions:

2ceV, the most critical reviewer who assigned a score of 3, notes that the comparisons against existing EA methods is not particularly fair (this concern is shared with other reviewers). 2ceV's opinion is that this work is not representative of evolutionary biology since the reward is fixed but not emerged, but on this point the Author's rebuttal is that their intention is to speed up computation, rather than mimic evolutionary biology (precisely because the emergence of reward via bi-level optimization is painfully slow), and I do think their rebuttal is valid here.

J3e1 is most supportive of the work, assigning a score of 6. J3e1 appreciates the proposed approach of providing alternatives to bi-level optimization. Their main concern is that the environments may not be open-ended, and also for the experiments, noted that the "authors pointed out that EvER works only in the MARL setting and we don't have a suitable bi-level baseline to compare to. To resolve this issue, I would suggest to either move to the single agent setting (if possible) or create an appropriate baseline." As the authors noted in the rebuttal, perhaps no baselines exist since the problem is infeasible for MARL systems. I believe this point should be better communicated in the narrative to manage expectations.

KvmB revised the score to 5 after several discussions with authors and other reviewers. Like J3e1, KvmB notes the novelty of the work and appreciates the direction and usefulness of the reward formulation to mimic gene selection to be promising. The main issue they have is the quality of the experiments, which falls short of expectations (several ideas and suggestions are given in the review and back and forth discussion). I also think that if the experiments were stronger, then this work will have a high impact.

In the end, even if we give the authors the benefit of discounting reviewer 2ceV (who does have a few valid points regarding EA comparisons), the work is still borderline (5 and 6), based on the reviews from J3e1 and KvmB which covers the main issues with the paper. I'm recommending a rejection of this work, but I do think it is a promising and interesting direction, especially as a way to avoid bi-level optimization, and I encourage the authors to take the inputs given to improve it further.